# Immune Response to Initial and Booster SARS-CoV-2 mRNA Vaccination in Patients Treated with Siponimod—Final Analysis of a Nonrandomized Controlled Clinical Trial (AMA-VACC)

**DOI:** 10.3390/vaccines11081374

**Published:** 2023-08-16

**Authors:** Tjalf Ziemssen, Marie Groth, Veronika Eva Winkelmann, Tobias Bopp

**Affiliations:** 1Department of Neurology, Center of Clinical Neuroscience, Carl Gustav Carus University Clinic, University Hospital of Dresden, Technische Universität Dresden, 01307 Dresden, Germany; 2Novartis Pharma GmbH, 90429 Nuremberg, Germany; marie.groth@novartis.com (M.G.); veronika.winkelmann@novartis.com (V.E.W.); 3Institute for Immunology, University Medical Center of the Johannes Gutenberg University, 55131 Mainz, Germany; boppt@uni-mainz.de

**Keywords:** COVID-19 vaccination, secondary progressive multiple sclerosis, disease-modifying therapy, neutralizing antibodies, T-cell response

## Abstract

Background: Evidence on SARS-CoV-2 mRNA vaccination under siponimod treatment is rare. Methods: AMA-VACC is a prospective, open-label clinical study on SARS-CoV-2 mRNA vaccination during ongoing siponimod treatment (cohort 1), during siponimod interruption (cohort 2), or during treatment with other disease-modifying therapies or without therapy (cohort 3). SARS-CoV-2-specific antibodies and T-cell reactivity were measured six months after the initial vaccination and one month after the booster. Results: 41 patients were recruited into cohort 1 (*n* = 17), cohort 2 (*n* = 4), and cohort 3 (*n* = 20). Seroconversion for SARS-CoV-2 neutralizing antibodies was reached by 50.0%, 100.0%, and 90.0% of patients at month 6 and by 81.3%, 100.0%, and 100.0% one month after booster (cohorts 1, 2, and 3, respectively). Antibody levels in cohort 1 increased after the booster compared to month 6 but remained lower compared to cohorts 2 and 3. T-cell responses were seen in 26.7%, 25.0%, and 73.7% at month 6 and in 28.6%, 50.0%, and 83.3% after the booster (cohorts 1, 2, and 3, respectively). In cohort 1, the extent of T-cell response was lower at month 6 compared to cohorts 2 and 3 but reached almost similar levels after the booster. Conclusions: The antibody and T-cell responses support SARS-CoV-2 (booster) vaccines in siponimod-treated patients.

## 1. Introduction

Vaccinations against SARS-CoV-2 have been instrumental in reducing the threat of the pandemic. They are now standard medical practice, and booster vaccinations are recommended especially for vulnerable individuals, including multiple sclerosis (MS) patients under immunomodulatory treatment. Initially, it was unclear to what extent people with secondary progressive MS (SPMS) receiving disease-modifying therapies (DMTs) would develop an effective immune response to the vaccines. It has been suggested that antibody response to vaccines is impaired in DMT-treated patients [1,2].

Siponimod is a DMT approved for use in patients with active SPMS. It inhibits lymphocyte sphingosine-1-phosphate (S1P)-1 and S1P5 receptors, leading to the retention of these cells in lymph nodes [3]. According to the summary of product characteristics (SmPC) for siponimod, temporary treatment interruption should be considered for the purpose of vaccination [4]. However, evidence on how patients treated with siponimod respond to the novel mRNA-based vaccines is rare, and it remains unclear whether it is necessary to interrupt treatment, which may increase the risk for disease progression. Many of the early studies on initial SARS-CoV-2 vaccination suggest seroconversion in most siponimod-treated patients [5,6,7]. However, antibody titers were significantly lower in patients treated with siponimod compared to healthy controls [5]. Previously published interim data from the AMA-VACC study have shown that approximately three quarters of patients developed SARS-CoV-2-specific humoral or cellular responses or both as soon as one week after full vaccination, irrespective of whether siponimod treatment was continued or interrupted [8]. Booster data are even rarer and include mixed S1P receptor (S1PR) inhibitor cohorts, with the majority of patients on fingolimod. Due to different pharmacokinetics and pharmacodynamics, data on other S1PR modulators cannot be transferred to siponimod [9,10].

Therefore, the AMA-VACC study investigated the cellular and humoral immune response to SARS-CoV-2 mRNA vaccines in SPMS patients treated with siponimod and vaccinated with and without treatment interruption, and in patients in a control group vaccinated while receiving other DMTs or no DMT. The results of the interim analysis, including the primary endpoint results on seroconversion rates, have been previously published [8]. These results are promising and have shown that most siponimod-treated patients mount relevant immune responses under continuous siponimod therapy. Interruption of treatment for the purpose of vaccination is not supported by initial AMA-VACC data [8].

However, the question remained open as to whether the immune response is maintained over several months and whether siponimod patients benefit from a booster vaccination, which is now recommended as standard. Therefore, we present the final study results, including evaluations of the immune response six months after initial vaccination and one month after booster vaccination. The results of AMA-VACC aim to guide treating physicians and patients in coordinating MS therapy and vaccination.

## 2. Materials and Methods

The study design as well as details on participants, treatments, assessments, endpoints, and statistical analyses have been published previously [8]. Briefly, AMA-VACC was a three-cohort, multicenter, open-label, prospective clinical study (EudraCT 2020-005752-38; NCT04792567) over twelve months in Germany. The study population consisted of patients with SPMS diagnoses and patients with relapsing-remitting MS (RRMS) at risk of developing SPMS. The initial SARS-CoV-2 mRNA vaccination cycle consisted of the first and second vaccinations; optional additional booster vaccinations were allowed at the physician’s discretion. Patients were treated with either siponimod, glatiramer acetate (GA), dimethyl fumarate (DMF), beta-interferons (IFN-beta), teriflunomide (TF), or no DMT as part of their clinical routine. Cohort 1 included patients vaccinated during ongoing siponimod treatment. Cohort 2 consisted of participants who interrupted siponimod therapy for the purpose of the initial vaccination. It was at the discretion of the physician and the patient to decide upon the preferred option, as both are in line with the SmPC of siponimod. Cohort 3 was designed as a control group. Patients in cohort 3 were vaccinated while receiving one of the other DMTs (DMF, GA, IFN-beta, or TF) or no DMT.

The trial was conducted in accordance with the International Conference on Harmonisation guidelines for Good Clinical Practice and the principles of the Declaration of Helsinki. The protocol was approved by the ethics committee “Technische Universität Dresden” (AMG ff-EK-34012021). All patients or their legal representatives provided written informed consent before commencing trial-related procedures.

Endpoints included the proportion of participants achieving seroconversion as defined by the detection of SARS-CoV-2 serum neutralizing antibodies (Nab), SARS-CoV-2 serum neutralizing antibody levels, SARS-CoV-2 total anti-spike antibody levels, and SARS-CoV-2-specific T-cell reactivity, measured as (i) the presence of SARS-CoV-2 reactive T-cells secreting either interferon-γ (IFN-γ) or interleukin-2 (IL-2) or both and (ii) IFN-γ secretion normalized for basal T-cell activity (IFN-γ stimulation indices). The cut-off for a positive T-cell response depended on the negative control to exclude a nonspecific increase in T-cell reactivity as a cause of increased IFN-γ secretion after stimulation with SARS-CoV-2 peptide. Assessments were performed one week, one month, and six months after the second dose of the vaccination cycle. An additional study visit was conducted one month after a booster vaccination. A final follow-up call to assess the occurrence of COVID-19 infections is intended 12 months after the second dose of the vaccination cycle.

The results of a pre-planned interim analysis, scheduled after all participants had completed the study visit one week after the second vaccination, constituted the primary analysis of the study and have been reported previously [8]. Here we report on the final study results, including month 6 data, antibody levels, and safety data.

No formal statistical testing was applied. All endpoints reported here were analyzed descriptively and presented as frequencies and percentages, mean and standard deviation (SD), or median and range. Statistical analyses were performed with SAS version 9.2, Cary, North Carolina, USA.

## 3. Results

Details on the study population have been reported previously [8]. Briefly, 41 patients were recruited (17 patients into cohort 1, four patients into cohort 2, and 20 patients into cohort 3). The median age of participants in the three cohorts was 51–56 years, 75 to 80% were female, and MS history was 9–18 years, with higher age and longer disease history in the siponimod cohorts (cohorts 1 and 2). All patients from cohorts 1 and 2 had SPMS. In cohort 3, the majority were RRMS patients (60%), while 10% had SPMS, and for 30%, the MS subtype was not specified. A booster vaccination was administered to 38 patients. Of these, 17 patients had received the booster before and 21 patients after month 6 (Table 1). The time since the start of siponimod treatment in cohort 1 and the duration of siponimod interruption for the initial vaccination in cohort 2 have also been published previously [8]. Accordingly, patients have been treated with siponimod for 0.63 years (median). In cohort 2, siponimod was stopped 15 days (mean) prior to the first vaccination and restarted 30 days (mean) after the second vaccination; the total duration of interruption was 77 days (mean). No information is available on the duration of the siponimod interruption for booster vaccination.

The endpoint of seroconversion for SARS-CoV-2 neutralizing antibodies at month 6 was reached by 50.0% of patients in cohort 1, 100.0% in cohort 2, and 90.0% in cohort 3. One month after booster vaccination, all but three patients in cohort 1 (81.3%) and all patients in cohorts 2 and 3 (100.0%) reached seroconversion. Overall, SARS-CoV-2 neutralizing antibody levels tended to be lower in cohort 1 compared to cohorts 2 and 3. In all cohorts, some patients tended to have lower SARS-CoV-2 neutralizing antibody levels at month 6 compared to month 1. Booster vaccination resulted in an increase in antibody levels compared to month 6 as well as compared to month 1 (Figure 1A). Levels of total anti-spike antibodies continuously increased with every vaccination in cohort 1, but overall, they remained below those in the other cohorts. In cohorts 2 and 3, total anti-spike antibody levels reached the detection limit in all but two patients at month 1 and in all patients at month 1 after booster (Figure 1B). A subgroup analysis by timing of booster vaccination (before or after month 6) showed similar results to the overall cohorts and indicated no relevant differences attributable to timing of vaccination (Appendix A).

SARS-CoV-2-specific T-cell responses were seen in only 26.7% (cohort 1) and 25.0% (cohort 2) at month 6, as well as in 28.6% (cohort 1) and 50.0% (cohort 2) at month 1 after the booster. On the contrary, most patients in cohort 3 showed T-cell responses at month 6 (73.7%) and one month after booster (83.3%). (Figure 2A). In cohort 1, however, the extent of T-cell response was lower at month 6 compared to cohorts 2 and 3 but increased after booster vaccination and reached almost similar levels. No marked difference in the extent of T-cell response was observed between cohorts 2 and 3, neither at month 6 nor one month after the booster (Figure 2B).

Overall, nine COVID-19 infections were reported: four in cohort 1 (all after the booster vaccination) and five in cohort 3 (four after the booster vaccination and one without the booster vaccination). All infections were mild except for one case of medium severity, and all patients fully recovered. The duration of infections was 7–16 days (Table 2).

Overall, adverse events (AEs) were reported by 29 patients (70.7%) during the study. AEs related to siponimod occurred in five patients (four had lymphopenia and two had increased liver function tests), and in 19 patients, AEs related to SARS-CoV-2 vaccines were reported. Serious adverse events were reported by one patient from each cohort. One patient from cohort 1 discontinued study medication (siponimod) due to adverse events (previously reported). No deaths occurred (Table 3). Two relapses occurred during the study, both more than five months after the last vaccination in cohort 1. These cases have been reported previously [8]. In cohorts 2 and 3, no relapses were observed.

## 4. Discussion

The AMA-VACC study is the first to systematically analyze both humoral and cellular immune responses to initial and booster SARS-CoV-2 vaccines in patients receiving siponimod. The final results show that the immune response persists after 6 months in patients vaccinated during continued treatment with siponimod, in patients vaccinated during siponimod interruption, and in patients receiving other DMT or no DMT at the time of their vaccination. While all patients vaccinated during siponimod interruption and all patients receiving other DMT or no DMT showed an adaptive immune response, the seroconversion rates on continuous siponimod treatment were only slightly lower, with over 80% of patients showing an adaptive immune response to SARS-CoV-2 mRNA vaccines one month after booster vaccination.

Evidence on booster vaccination in siponimod-treated patients is still rare. However, the results on humoral responses are in line with early studies on the initial SARS-CoV-2 vaccination. Accordingly, seroconversion rates reported after initial vaccination in siponimod-treated patients ranged from 80 to 88% [5,6,7], compared to 100% in healthy controls [5]. Antibody titers were significantly lower in patients receiving siponimod than in healthy controls [5]. These early studies focused on humoral responses. However, in addition to the development of antibodies, the T-cell response plays a fundamental role after SARS-CoV-2 vaccination. Initially, cytotoxic CD8^+^ T-cells are activated, while the secondary response involves CD4^+^ T-cell proliferation, leading to T-helper-2 (Th2)-mediated B-cell stimulation and T-helper-1 (Th1)-mediated cytotoxicity [11,12]. Assessing both antibody-dependent and T-cell-mediated responses after mRNA vaccination is therefore indicated in patients treated with siponimod, which induces lymphocyte retention through inhibition of S1P1 and S1P5 receptors on lymphocytes [3]. Data on the immune response, including T-cell reactivity, after the initial vaccination were available from interim data from the AMA-VACC study. Accordingly, over 70% of patients continuously treated with siponimod and 75% with an interruption of siponimod treatment developed SARS-CoV-2-specific humoral or cellular responses or both as soon as one week after full vaccination [8]. Available booster vaccination data usually include mixed cohorts receiving different S1PR inhibitors. Accordingly, results on booster vaccination (i.e., third vaccination) are available for eight patients from a cohort study that consisted of eleven patients on fingolimod and two patients on siponimod. However, it remains unclear whether the booster data also includes siponimod-treated patients. At a mean time of 33 days after the third vaccination, seroconversion for total anti-spike antibodies was detected in 100% of patients, and 75% of patients had antibodies against the spike receptor-binding domain. Spike-specific CD4^+^ and CD8^+^ T-cell responses were reported to be maintained following the third vaccination [13]. Another observational study, including two patients on siponimod, eight patients on fingolimod, and two patients on ozanimod, measured immune responses approximately four weeks after the third vaccination. Neutralizing antibodies against wildtype SARS-CoV-2 were detected in approximately half of patients, and only one quarter of patients had neutralizing antibodies against the omicron variant of SARS-CoV-2. Spike-specific T-cell reactivity was not significantly increased by a booster vaccination [14]. Due to differences in pharmacokinetics and pharmacodynamics, data generated in patients receiving other S1PR modulators should not be transferred to siponimod [9]. In this context, a published literature review by Baker et al. indicates that the newer generation of S1PR modulators is advantageous over the older generation regarding the response to SARS-CoV-2 vaccination [15]. Accordingly, the authors suggested stronger vaccination responses under siponimod, ozanimod, and ponesimod compared to fingolimod [15]. Conversion rates of 60 to 77% have been reported for fingolimod [15]. Seroconversion rates reported after initial vaccination in siponimod-treated patients ranged from 80 to 88%, as outlined above [5,6,7]. It was hypothesized by Baker et al. that the S1PR4-sparing effect of newer generation S1PR modulators might be beneficial in the context of vaccination response [15]. However, it has to be pointed out that data on SARS-CoV-2 vaccination response in patients treated with S1PR modulators other than fingolimod are rare, even in studies including several S1PR modulators [13]. Comparison of seroconversion rates and T-cell response levels for different S1PR modulators is therefore difficult.

One major question in the context of vaccination for patients receiving DMTs is whether temporary discontinuation of treatment is necessary to allow for effective immunization. The SmPC for siponimod recommends considering treatment interruption one week prior to vaccination until four weeks after vaccination [4]. AMA-VACC shows relevant immune reactivity towards SARS-CoV-2 vaccination during continued siponimod treatment. On the contrary, the results generated during the siponimod interruption are not substantial enough to serve as guidance. However, given that an immune response can be initially achieved and boosted under continued siponimod treatment, the present results at least suggest that treatment interruption might be dispensed with. Furthermore, during AMA-VACC, no severe COVID-19 cases in patients vaccinated with continuous siponimod have been reported, which indicates effective prevention of severe disease courses.

The current data from AMA-VACC are not suitable to make a statement regarding T-cell reactivity after a booster. As previously shown, T-cell response in siponimod-treated patients peaked early after vaccination [8], and according to the present data, it increased only slightly after a booster vaccination, while it steadily increased in the control group. Unfortunately, the response to the booster vaccination has only been assessed one month after the booster, and it can be hypothesized that the T-cell response was not adequately captured. Furthermore, as outlined previously [8], the meaningfulness of T-cell assays in siponimod-treated patients is potentially limited by reduced numbers of circulating T-cells, and consequently, the T-cell response in this cohort might be underestimated. This is a result of the mode of action of siponimod, which reduces the proportion of circulating CD3^+^ T-lymphocytes and thereby the number of T-cells in the assays. Nevertheless, the development of neutralizing antibodies suggests functional T-cell-B-cell interaction in all patients.

Despite these encouraging results, the present study has some limitations. First, the sample size in the study is very small. In particular, the sample size of the siponimod interruption cohort (cohort 2), which includes only four patients, is too small to draw valid conclusions regarding the necessity of treatment interruption for vaccination. Therefore, overall, the present results need to be confirmed in further studies. Nevertheless, the results of AMA-VACC allow for the assumption that an immune response towards SARS-CoV-2 mRNA vaccines is elicited in siponimod-treated patients. As the decision to continue or interrupt siponimod treatment for the purpose of vaccination was at the discretion of the physician, the small sample size in cohort 2 might reflect a reluctance to interrupt disease-modifying treatment in MS patients. This highlights the need for data on vaccination during continued siponimod treatment for well-informed treatment decisions. Second, the timing of the booster vaccination in AMA-VACC was not in line with current recommendations. The reason for this discrepancy is that study recruitment had already been initiated before the Robert Koch-Institute (RKI) recommended a third vaccination in severely immunocompromised patients four weeks after the second vaccination [16]. Approximately half a year after the start of recruitment, the Deutsche Multiple Sklerose Gesellschaft (DMSG), the Kompetenznetz Multiple Sklerose (KKNMS), and the Berufsverband Deutscher Neurologen (BDN) issued a joint statement to clarify that the RKI recommendation should also be applied to MS patients treated with anti-CD20 antibodies or S1P-inhibitors [17]. This means that in AMA-VACC, the booster vaccination was performed later than currently recommended. Of note, in AMA-VACC, no difference was observed between patients who received their booster before or after month 6. Third, as already discussed [8], participants in the control cohort are younger and have a shorter MS history than patients in the siponimod cohorts. This might impact the analysis, as higher age has been shown to be negatively correlated with SARS-CoV-2 neutralizing antibody titers after vaccination [18,19]. 

In summary, the results of the final AMA-VACC analysis show that patients receiving continuous siponimod treatment can mount humoral and cellular immune responses, albeit somewhat less pronounced than patients receiving other DMT or no DMT. Booster vaccination increases antibody levels and T-cell reactivity in patients continuously treated with siponimod. A recommendation towards or against siponimod treatment interruption for the purpose of vaccination cannot be given based on AMA-VACC data. However, the study results support booster vaccination of siponimod-treated patients. 

## Figures and Tables

**Figure 1 vaccines-11-01374-f001:**
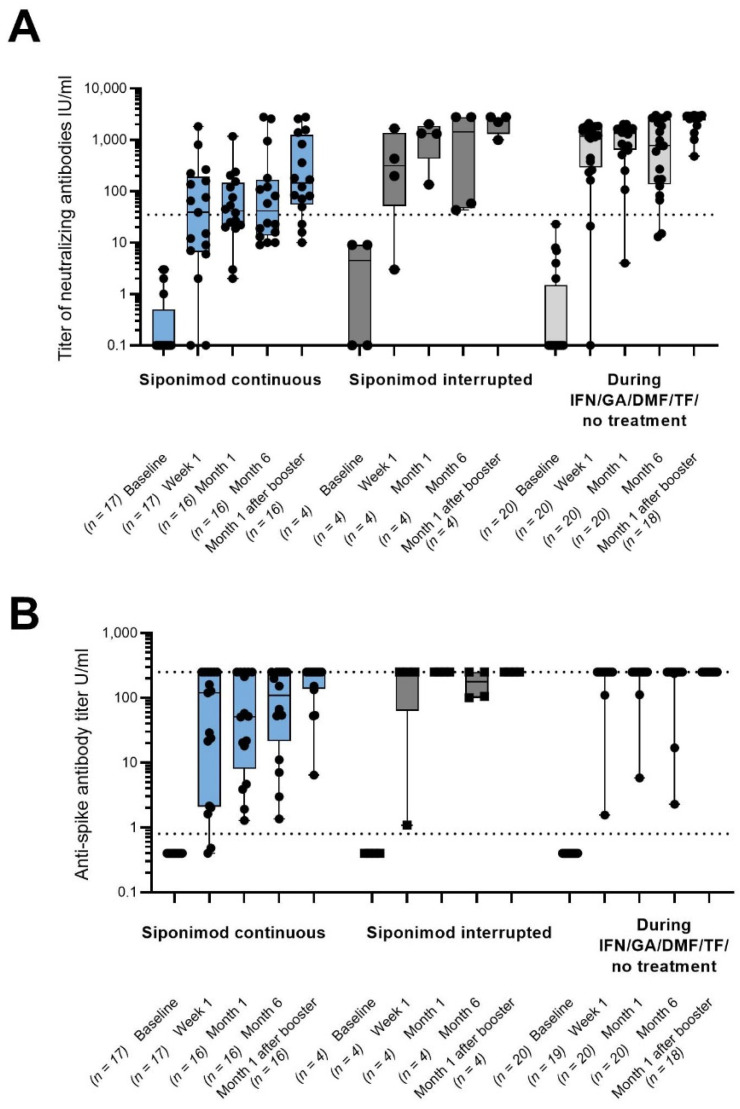
(**A**) SARS-CoV-2-specific neutralizing antibody levels in U/mL. (**B**) SARS-CoV-2-specific serum total antibody levels in U/mL. All the patients with available data were included in the analysis, and individual values are represented by dots. For 11 booster patients, the month 6 visit and the month 1 after the booster visit were identical (cohort 1: n = 7; cohort 2: n = 1; cohort 3: n = 3). The bars show the median values; the black dotted lines indicate assay-specific cut-offs for seropositivity; and the gray dotted lines indicate the maximal value of the quantification range. DMF: dimethyl fumarate; GA: glatiramer acetate; IFN: interferon-beta; n: number of patients with assessments; TF: teriflunomide; and U: units.

**Figure 2 vaccines-11-01374-f002:**
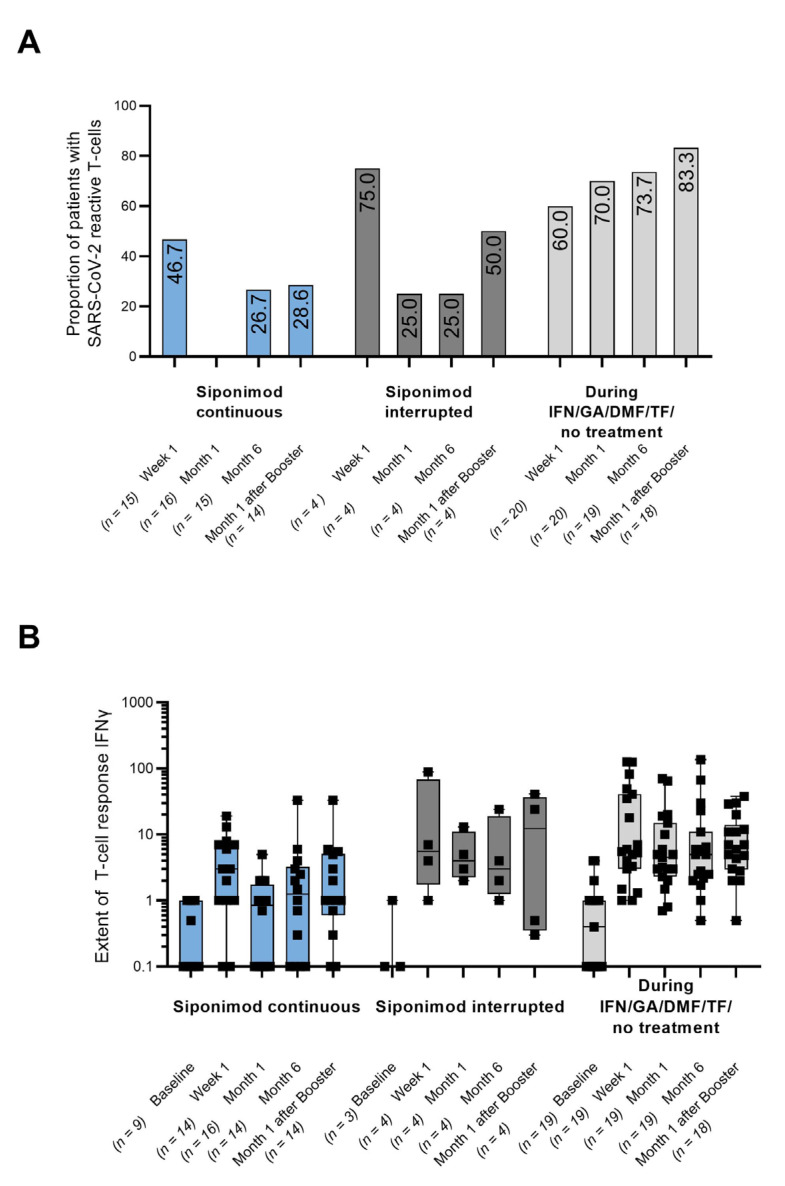
(**A**) T-cell response defined as the presence of SARS-CoV-2-reactive T-cells measured by the secretion of either IFN-, IL-2, or both (any level above basal activity); (**B**) ELISpot-based quantification of T-cell reactivity by calculation of IFN- stimulation indices towards SARS-CoV-2. Each dot represents one patient, and the medians are indicated by horizontal lines. DMF: dimethyl fumarate; GA: glatiramer acetate; IFN: interferon-beta; IFN-: interferon-; n: number of patients with assessments; PBMC: peripheral blood mononuclear cell; and TF: teriflunomide. The T-cell response could not be assessed in three patients with the continued siponimod treatment, one patient in the control group at the month 6 visit, and two patients of cohort 3 at month 1 after the booster because of insufficient cell counts after PBMC isolation. For 11 booster patients, the month 6 visit and the month 1 after the booster visit were identical (cohort 1: n = 7; cohort 2: n = 1; cohort 3: n = 3).

**Table 1 vaccines-11-01374-t001:** Booster vaccination characteristics.

Variable *	Cohort 1 Siponimod Continuous (*N* = 17)	Cohort 2 Siponimod Interrupted (*N* = 4)	Cohort 3 DMF/GA/IFN/TF/No DMT (*N* = 20)
*N*	17	4	20
Booster vaccination			
Before visit 3 (month 6)	8 (47.1)	1 (25.0)	8 (40.0)
After visit 3 (month 6)	8 (47.1)	3 (75.0)	10 (50.0)
No booster vaccination	1 (5.9)	0 (0.0)	2 (10.0)
Type of booster vaccination, *n* (%)			
BioNTech/Pfizer|Moderna	11 (64.7)|5 (29.4)	2 (50.0)|2 (50.0)	11 (55.0)|7 (35.0)
Cross vaccinations	4 (23.5)	2 (50.0)	6 (30.0)
Vaccination time interval			
Second vaccination to booster, months	5.74 [4.95; 7.41]	6.44 [5.38; 6.89]	5.82 [5.18; 6.52]
Booster to visit 3 (month 6) for patients with booster before visit 3, days; mean (SD)	*N*′ = 8 21.0 ± 7.47	*N*′ = 1 21.0 ± 0	*N*′ = 8 13.5 ± 9.38

* If not indicated otherwise, data are presented as median (min; max). DMF: dimethyl fumarate; DMT: disease-modifying treatment; GA: glatiramer acetate; IFN: interferon-beta; *N*: number of patients in population; *N*′: number of patients in subpopulation; *n*: number of patients in category; SD: standard deviation; TF: teriflunomide. * No cross-vaccination was documented for 26 of 38 booster vaccinations. All 12 cross-vaccinations consisted of an initial vaccination with BioNTech/Pfizer and a booster vaccination with Moderna.

**Table 2 vaccines-11-01374-t002:** COVID-19 cases reported in the study.

Variable *	Cohort 1 Siponimod Continuous (*N* = 17)	Cohort 2 Siponimod Interrupted (*N* = 4)	Cohort 3 DMF/GA/IFN/TF/No DMT (*N* = 20)
Patients with a COVID-19 infection	*N*′ = 4	*N*′ = 0	*N*′ = 5
Median duration of infection (days)	12 [10;17]	n/a	10 [8;11]
CTCAE grade, *n* (%)			
mild	3 (75.0)	n/a	5 (100.0)
moderate	1 (25.0)		0 (0.0)
severe or higher	0 (0.0)		0 (0.0)
Fully recovered, *n* (%)	4 (100.0)	n/a	5 (100.0)
Treatment interruption necessary, *n* (%)			
yes	0	n/a	0 (0.0)
no	4 (100.0)		5 (100.0)

* If not indicated otherwise, data are presented as median (min; max). CTCAE: Common Terminology Criteria for Adverse Events; DMF: dimethyl fumarate; DMT: disease-modifying therapy; GA: glatiramer acetate; IFN: interferon-beta; *N*: number of patients in population; *N*′: number of patients in subpopulation; *n*: number of patients in category; TF: teriflunomide.

**Table 3 vaccines-11-01374-t003:** Overview of adverse events.

Adverse Events (AEs), *n* (%)	Cohort 1 Siponimod Continuous (*N* = 17)	Cohort 2 Siponimod Interrupted (*N* = 4)	Cohort 3 DMF/GA/IFN/TF/No DMT (*N* = 20)
Any AEs	10 (58.8)	3 (75.0)	16 (80.0)
AEs by SOC			
General disorders and administration site conditions	4 (23.5)	3 (75.0)	7 (35.0) ^b^
Nervous system disorders	4 (23.5)	2 (50.0)	5 (25.0)
Musculoskeletal and connective tissue disorders	4 (23.5)	1 (25.0)	3 (15.0)
Investigations	2 (11.8)	2 (50.0)	2 (10.0)
Blood and lymphatic system disorders	3 (17.6)	1 (25.0)	0
Infections and infestations	7 (41.2)	1 (25.0)	6 (30.0)
Ear and labyrinth disorders	0	0	1 (5.0)
Eye disorders	1 (5.9)	0	0
Gastrointestinal disorders	0	0	1 (5.0)
Hepatobiliary disorders	0	0	1 (5.0)
Immune system disorders	0	0	1 (5.0)
Injury, poisoning, and procedural complications	1 (5.9)	0	0
Psychiatric disorders	0	1 (25.0)	0
Reproductive system and breast disorders	1 (5.9)	0	0
Respiratory, thoracic, and mediastinal disorders	1 (5.9)	0	0
Vascular disorders	1 (5.9)	1 (25.0)	0
Not coded ^a^	1 (5.9) ^a^	0	0
AEs related to study medication	8 (47.1)	2 (50.0)	10 (50.0)
Any SAEs	1 (5.9)	1 (25.0)	1 (5.0)
SAEs by SOC and *PT*			
Infections and infestations	1 (5.9)		1 (5.0)
*Escherichia urinary tract infection*	1 (5.9)		
*Acute sinusitis*			1 (5.0)
*Gastroenteritis rotavirus*			1 (5.0)
Nervous system disorder			
*Epilepsy*		1 (25.0)	

In the case of multiple AEs, a patient is counted only once in the respective category. AE: adverse event; DMF: dimethyl fumarate; DMT: disease-modifying therapy; GA: glatiramer acetate; IFN: interferon-beta; *N*: number of patients in population; *n*: number of patients in category; PT: preferred term; SOC: system organ class; TF: teriflunomide. ^a^: hospitalization. ^b^: One event has been recoded for the final analysis (coding in interim analysis: PT vaccination site reaction; SOC General disorders and administration site conditions in the interim analysis; final analysis: PT immunization reaction; SOC immune system disorders).

## Data Availability

Data will be provided upon reasonable request.

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
