# Peer review of "Immune Response to Initial and Booster SARS-CoV-2 mRNA Vaccination in Patients Treated with Siponimod—Final Analysis of a Nonrandomized Controlled Clinical Trial (AMA-VACC)"

_vaccines, 2023, doi:10.3390/vaccines11081374_

Round 1

Reviewer 1 Report

The study is interesting but some issues should be addressed.

1. Clinical and epidemiological data of each cohort (Including age, sex, disease duration, MS type, EDSS score), should be included and differences commented.

2. Reason to consider T cell response positive ore negative should be explained.

3. A cut-off for IFN-gamma production by T cells should be established.

4. Statistical analyses at least for cohorts 1 and 3 should be performed.

Author Response

We thank the reviewer for the thorough evaluation of our manuscript. We have carefully considered and responded to each suggestion. Please find our separate point-by-point response below. 

Comments to the Author

The study is interesting but some issues should be addressed.

1) Clinical and epidemiological data of each cohort (Including age, sex, disease duration, MS type, EDSS score), should be included and differences commented

Response: Clinical and epidemiological data per cohort have been published together with the interim analysis of month 1 data after the initial vaccination (Ziemssen et al. 2022; doi:10.1177/17562864221135305). To avoid duplicate publication, we have referenced the primary publication and briefly described the key characteristics in the main text. This paragraph already included age and disease duration. We have revised this section to include sex and MS type as well. Unfortunately, EDSS score was not collected in the study (lines 115 to 119).

2) Reason to consider T cell response positive ore negative should be explained;

Response: We thank the reviewer for this comment. We have added a paragraph on the role of T-cells in the response towards SARS-CoV-2 mRNA vaccines to explain why T-cell response has been considered in the AMA-VACC study (lines 225 to 231).

3) A cut-off for IFN-gamma production by T cells should be established.

Response: We understand from the reviewer’s comment that our approach on measuring T-cell activation was not explained clearly enough. We therefore like to elaborate on the T-cell response assessments in our manuscript. Baseline IFN-γ and IL-2 production can vary substantially between patients. To account for background response, specific SARS-CoV-2 T-cell reactivity was normalized for basal T-cell activity. The cut-off for a positive T-cell response depended on the negative control to exclude nonspecific increase in T-cell reactivity as a cause of increased IFN-γ secretion after stimulation with SARS-CoV-2 peptide. The overall T-cell response was defined as presence of SARS-CoV-2 reactive T-cells secreting either IFN-γ or IL-2 or both. The extent of T-cell response was defined as T-cell reactivity normalized for basal T-cell activity and measured by IFN-γ secretion (IFN-γ stimulation indices). We have revised the methods section to include an explanation of this approach (lines 98 to 100).

4) Statistical analyses at least for cohorts 1 and 3 should be performed.

Response: We agree that statistical analyses would help interpretation. However, the AMA-VACC trial was designed as a descriptive study, thus, no formal statistical testing was applied, and all endpoints were analyzed descriptively. Comparison of cohorts 1 and 3 was also planned descriptively according to the study protocol of the AMA-VACC trial. Post-hoc statistical analyses have not been included as these could easily be subject to misinterpretation especially in small cohorts.

Reviewer 2 Report

1. Were n=5 Covid infections in cohort 2 or 3? (See Table 2 and text page 6, lines 162-163)

2. I think reporting humoral or cellular vaccine response is a bit misleading, since you want both. I would report each solo.

3. How long were the Group 2 (n=4) people off siponimod?

4. How long were people on siponimod in Group 1?

5. The data presented supports that Group 1 had lower responses to vaccination on a humoral and cellular basis. Do the authors not agree? They should state that clearly.

6. Please have a more robust discussion of vaccine response in second generation vs. fingolimod-treated MS, to put this into a more valuable context. 

7. Group 2 is very limited with n=4; please clearly acknowledge.

8. Did the authors measure the T-cell response consistent with prior other studies on S1P-R modulators? 

Author Response

We thank the reviewer for the thorough evaluation of our manuscript. We have carefully considered and responded to each suggestion. Please find our separate point-by-point response below.

Comments to the Author

1) Were n=5 Covid infections in cohort 2 or 3? (See Table 2 and text page 6, lines 162-163).

Response: We thank the reviewer for highlighting our mistake in the text. In cohort 2, none of the patients had COVID, while 5 patients in cohort 3 reported COVID. We have revised the text to correct for this error (line 179).

2) I think reporting humoral or cellular vaccine response is a bit misleading, since you want both. I would report each solo

Response: We agree with the reviewer that both aspects of the immune response are important and should be considered separately, which is the case in our present paper. The initial paper on month 1 data additionally included a conjoint analysis on the number of patients having either humoral or cellular response or both, which is referred to in the introduction of the present paper.

3) How long were the Group 2 (n=4) people off siponimod?

Response: We thank the reviewer for this important question. As described in the methods section, cohort 2 consisted of participants who interrupted siponimod therapy for the purpose of the initial vaccination. The time off siponimod was reported together with the results of month 1 data after the initial vaccination (Ziemssen et al. 2022; doi:10.1177/17562864221135305). Accordingly, siponimod was stopped 15 days (mean) prior to the first vaccination and restarted 30 days (mean) after the second vaccination; the total duration of interruption was 77 days (mean). As booster vaccination was optional, unfortunately, we have no data on whether and how long siponimod was paused for booster vaccination. We have revised the results section to include information on the duration of siponimod interruption for the initial vaccination (lines 120 to 127).

4) How long were people on siponimod in Group 1?

Response: We agree that the time on siponimod treatment is an important information. Time on current treatment (i.e., siponimod) has therefore been reported previously together with our results on month 1 after initial vaccination (Ziemssen et al. 2022; doi:10.1177/17562864221135305). Accordingly, patients in cohort 1 have been on their current treatment for 0.63 years (median). We have included this information in the revised results section (lines 120 to 127).

5) The data presented supports that Group 1 had lower responses to vaccination on a humoral and cellular basis. Do the authors not agree? They should state that clearly

Response: We agree with the reviewer regarding the overall response in cohort 1. We have revised the results section and the discussion section to include clear statements on this observation were these have not yet been included (lines 139 to 140; 145; 211 to 219). However, regarding the extent of T-cell response, to our interpretation, cohort 1 reaches at least almost similar levels as cohort 2 and 3, which clearly supports the benefit of a booster in patients vaccinated during ongoing siponimod (Figure 2B).

6) Please have a more robust discussion of vaccine response in second generation vs. fingolimod-treated MS, to put this into a more valuable context

Response: We agree that second generation S1PR vs. fingolimod in the context of SARS-CoV-2 vaccination is an interesting topic that deserves discussion. We have revised the discussion section to expand the paragraph (lines 256 to 265). An extensive discussion and comparative elaboration of vaccination response under different S1PR modulators, however, is difficult, especially since few data are available from other S1PR modulators than fingolimod. In our opinion, this lack underlines the urgent need for data on newer generation S1PR modulators such as siponimod.

7) Group 2 is very limited with n=4; please clearly acknowledge

Response: We agree that the sample size was small, especially in cohort 2, and results require further confirmation. We have already mentioned this limitation in the discussion section but have now further elaborated on this topic, as we think that the small numbers in this group might also reflect reluctance of treating physicians to interrupt disease-modifying treatment for the purpose of vaccination (lines 299 to 303). Despite the small sample sizes and the uncertainties this brings, we are convinced that the present results at least allow to assume that an immune response is elicited under siponimod treatment. Assuming reluctance to interrupt treatment, these data are urgently needed to inform treatment decisions in MS care and support vaccination during continuous siponimod treatment.

8) Did the authors measure the T-cell response consistent with prior other studies on S1P-R modulators?

Response: IFN-gamma release upon stimulation is a common method of assessing T-cell activation. To our knowledge, this is the method mostly used in studies assessing vaccination response under S1PR modulators. Accordingly, T-cell response was assessed by IFN-gamma release upon stimulation in studies by, e.g., Akgün et al. 2022, Multiple Sclerosis Journal; 28: (3S) 347; Tortorella et al. 2022, doi:10.1212/WNL.0000000000013108. Another method to assess T-cell response is by measuring activation-induced markers, which was used by Sabatino et al 2023, doi: 10.1016/j.msard.2022.104484.

Round 2

Reviewer 1 Report

The authors addressed adequately all my concerns.

Author Response

We thank the reviewer for thorough evaluation of our manuscript. 

Reviewer 2 Report

In the second paragraph page 1 (lines 39-40) you might want to add for clarity; ".....by retaining these cells in lymph nodes"  (e.g., not receptors)

Author Response

We thank the reviewer for thorough evaluation of our manuscript. We have revised the sentence for more clarity (revised version: It inhibits lymphocyte S1P1 and S1P5 receptors leading to retention of these cells in lymph nodes).